# Identification of TRPM2 as a Potential Therapeutic Target Associated with Immune Infiltration: A Comprehensive Pan-Cancer Analysis and Experimental Verification in Ovarian Cancer

**DOI:** 10.3390/ijms241511912

**Published:** 2023-07-25

**Authors:** Danxi Zheng, Siyu Long, Mingrong Xi

**Affiliations:** 1Department of Gynecology and Obstetrics, West China Second University Hospital, Sichuan University, No. 20, Third Section of People’s South Road, Chengdu 610041, China; 2015181622064@stu.scu.edu.cn; 2Key Laboratory of Birth Defects and Related Diseases of Women and Children, Sichuan University, Chengdu 610041, China; 3Laboratory of Molecular Translational Medicine, Center for Translational Medicine, Key Laboratory of Birth Defects and Related Diseases of Women and Children, West China Second Hospital, Sichuan University, Chengdu 610041, China; siyulong0109@alu.scu.edu.cn

**Keywords:** TRPM2, pan-cancer, immune infiltration, tumor-associated macrophages, ovarian cancer, targeted therapy, immunotherapy

## Abstract

The exact role of Transient receptor potential melastatin 2 (TRPM2) in tumor progression and immunomodulation remains elusive. We comprehensively investigated the expression pattern, diagnostic value, prognostic impact, genetic and epigenetic alterations of TRPM2 in pan-cancer. Then, we explored underlying pathways associated with TRPM2 and immune-related signatures. Ovarian cancer (OV) specimens were enrolled to test the expression of TRPM2 by immunohistochemistry and RT-qPCR. OV cell A2780 transfected with shRNA targeting TRPM2 was used in subsequent experiments. TRPM2 was aberrantly expressed and associated with unfavorable prognosis across various cancers. It possesses significant diagnostic values with AUC > 0.90. TRPM2 participated in pathways mediating immunoregulation and tumorigenesis. The expression of TRPM2 was significantly correlated with tumor microenvironment scores, tumor-stemness index, macrophages infiltration, immune checkpoints, and immune-related genes. OV single-cell datasets also indicated that TRPM2 was predominantly distributed on macrophages and malignancies. The overexpressed TRPM2 in OV tissues was validated at both the mRNA and protein levels. TRPM2 expression was significantly correlated with type2 macrophage marker CD206. Knockdown of TRPM2 inhibited OV cell proliferation and promoted apoptosis. Overall, TRPM2 has relevance to an immunosuppressive tumor microenvironment by modulating macrophage. It could serve as a powerful biomarker for tumor screening and prognosis, and a potential therapeutic target for tumor treatment, especially for OV.

## 1. Introduction

Nowadays, the incidence of cancer is increasing and cancer has become a leading cause of death worldwide, which imposes an enormous health and economic burden on the global healthcare system [1]. The successful clinical application of immunotherapy in some cancer types has subverted the landscape of traditional cancer treatment and Immunotherapy with immune checkpoint inhibitors (ICIs) constitutes a milestone in the advancement of cancer therapy [2,3]. However, only a minority of patients benefit from these novel therapeutic regimens. One major hurdle to the efficacy of any targeted immunotherapy is the highly immune dysfunctional tumor microenvironment (TME) [4]. The tumor microenvironment refers to the surrounding environment where tumors grow, containing diverse cell types, such as stromal and tumor-associated immune cells. Tumor cells and their microenvironment mutually influence each other: the signals from the tumor microenvironment have tumor-promoting and tumor-suppressing effects. Conversely, tumor cell-intrinsic factors can reshape the nature of the tumor immune microenvironment [5]. Hence, TME disorder plays crucial roles in tumor sustained growth and metastasis, and influences the outcome of immunotherapy [6]. Additionally, an increasing number of studies have revealed that the altered gene expression and accumulation of mutations in oncogenes and/or suppressors modulate the surrounding tumor microenvironment and host immune responses, ultimately facilitating cancer malignant progression [7]. Therefore, finding novel molecular targets to reverse the immunosuppressive microenvironment and halt or slow tumor progression is an urgent priority.

Transient receptor potential (TRP) channels are essential for cellular ion (mainly Ca^2+^ and Na^+^) transport and homeostasis [8], which takes part in a diverse array of biological processes. The TRPM subfamily is a subgroup of the TRP channel superfamily, encompassing TRPM1-TRPM8 [9]. Of these, Transient receptor potential melastatin 2 (TRPM2) functions as a calcium (Ca^2+^)-permeable, nonselective cation channel widely expressed in many cell types [10]. Intracellular reactive oxygen species (ROS), Ca^2+^ concentration, hydrogen peroxide, and adenosine diphosphate ribose (ADPR) can activate human TRPM2 [11,12,13]. Activation of TRPM2 promotes the expression of transcription factors and kinases which are essential for cell proliferation and survival, including HIF-1/2α, nuclear factor-related factor-2 (Nrf2), and Src phosphorylation [14]. Recent studies have demonstrated that TRPM2 is implicated in a variety of human complex diseases, including various types of malignancies. For instance, activation of TRPM2 not only enhances cell proliferative ability but also decreases sensitivity to chemotherapy in neuroblastomas. Down-regulation of TRPM2 expression or function contributes to decreased cancer cell proliferation and survival in many malignancies, such as breast, gastric, pancreatic, neuroblastoma, T-cell and acute myelogenous leukemia, and clear cell renal cell carcinoma [15,16,17,18,19]. An up-to-date review highlights the pro-tumorigenic behaviors of TRPM2 in several cancers and proposes targeting TRPM2 mediated Ca^2+^ signaling as a feasible anti-cancer therapy strategy [20]. Contrarily, there are also reports documenting that the activated TRPM2 induced the increase of apoptosis in the colon and breast cancer cells [21,22]. Additionally, although the expression of TRPM2 has been detected in many immune cell types [23], the function of TRPM2 in anti-tumor immunity has remained elusive. Thus, it is exceptionally valuable to explore more deeply the molecular mechanisms and immunoregulatory roles of TRPM2 in the pan-cancer dataset, so that new strategies can be developed for precision cancer therapy.

In the present study, we utilized resources from multiple databases to perform a comprehensive pan-cancer analysis for TRPM2 to investigate its molecular characteristics, diagnostic and prognostic value. Functional and pathway enrichment analyses were carried out to determine the potential function of TRPM2 in tumorigenesis. Moreover, we evaluated the association between TRPM2 expression and immune-related biomarkers such as immune cell infiltration and checkpoint-related genes to further explore the relevance of its expression to tumor immunity. Drug sensitivity data were also analyzed to better evaluate its value in the direction of tumor immunotherapy. Finally, we verified the bioinformatic results using the clinic specimens and cell line of ovarian cancer. Our research provided more information for a better understanding of the significance of TRPM2 in various cancers, highlighting the potential of TRPM2 as a therapeutic target for ovarian cancer treatment.

## 2. Results

### 2.1. The TRPM2 mRNA Expression Profile across Pan-Cancer

Considering that there were no corresponding normal tissue data available for some tumor types in The Cancer Genome Atlas (TCGA), we combined the data from Genotype-Tissue Expression (GTEx) and TCGA databases to investigate the expression level of TRPM2 in 33 cancer types. The results showed that TRPM2 was generally more highly expressed in cancer tissues compared to the corresponding normal tissues. On the other hand, a significant downregulation of TRPM2 expression was found in ACC, GBM, and LGG, and no statistically significant change was observed in the case of KICH, PCPG, THCA, and THYM (Figure 1A. Please see the complete list of the TCGA cancer-type abbreviations in Appendix A). The relative expression of TRPM2 in TCGA pan-cancer is displayed in Figure 1B, with the highest expression in LAML. As for TCGA paired analysis, TRPM2 was found to be highly expressed in 18 tumor types, including BLCA, BRCA, CHOL, COAD, ESCA, HNSC, KICH, KIRC, KIRP, LIHC, LUAD, LUSC, PRAD, READ, STAD, THCA, and UCEC (Figure 1C). The result from the Cancer Cell Line Encyclopedia (CCLE) database also suggested that the expression level of TRPM2 was relatively high in LAML, PRAD, ALL, UCEC, ESCA, LCML, COAD/READ, OV, and BRCA tumor cells (Figure 1D).

### 2.2. TRPM2 Alteration Analysis across Various Cancer Types

The genetic alterations of TRPM2 in various cancers from the TCGA database were obtained. As demonstrated in Figure 2A, the most common alteration type was gene “Mutation,” followed by “Deep Deletion”, and “Amplification”. The mutation of TRPM2 most commonly occurred in melanoma, with mutation frequency up to almost 15%. Subsequently, we explored the relationship between TRPM2 expression and genomic variation in pan-cancer through integrating copy number alteration (CNA) and gene expression data. The results revealed that CNA had a significant positive correlation with TRPM2 expression in OV, UCEC, BRCA, GBM, CESC, HNSC, BLCA, LUSC, and KIRC (Figure 2B). Considering that promoter methylation also results in abnormal expression of genes, we further investigated the relationship between DNA promoter methylation level and TRPM2 expression among pan-cancer. The results suggested that DNA hypermethylation exerted a negative association with the expression level of TRPM2 in almost all cancer types, especially in PRAD and COAD (Figure 2C). Moreover, some well-known oncogenes and tumor suppressor genes showed dramatically high mutational frequencies in high TRPM2 expression groups. For instance, TP53 showed the highest mutation rate of 86% in UCS, followed by OV (76%), and LUSC (68%) (Appendix A).

### 2.3. Prognostic and Diagnostic Value of TRPM2 in Various Cancers

The above analysis clarified that TRPM2 was significantly differentially expressed among 27 cancer types. We performed Kaplan–Meier (KM) analysis in pan-cancer, and univariate Cox regression analysis to further assess the survival association between TRPM2 expression and patient prognosis in different tumors. The results suggested that the patients with a high TRPM2 expression had shorter survival times compared with those with low TRPM2 expression in ACC, BRCA, GBM, HNSC, KIRC, LAML, LGG, LIHC, LUSC, OV, STAD, TGCT, THYM, UCS, and UVM (Figure 3A). Univariate Cox analysis showed that TRPM2 expression was associated with overall survival (OS, *p* < 0.05) in BRCA, GBM, KIRC, LGG, LIHC, THYM, and UVM (Figure 3B); with disease-specific survival (DSS, *p* < 0.05) in BRCA, GBM, KIRC, LGG, THYM, and UVM; with disease-free interval (DFI, *p* < 0.05) in BRCA; with progression-free interval (PFI, *p* < 0.05) in BRCA, GBM, KIRC, LGG, THYM, and UVM (Appendix A), while enhanced TRPM2 expression exerted a positive impact on survival in patients with SKCM and THCA. Overall, we observed that TRPM2 expression was strongly associated with the survival outcomes of patients with multiple tumor types. Subsequently, the TCGA database was used to analyze the relationship between TRPM2 RNA expression and tumor clinical stage. As displayed in Figure 3C, the expression levels of TRPM2 were significantly different as stage increased in BRCA, KIRC, LUSC, PAAD, TGCT, and DLBC. We further conducted Receiver operating characteristic (ROC) curve analysis to investigate the possible diagnosis value of TRPM2 across pan-cancer. Results showed that TRPM2 had good diagnostic capabilities (area under the receiver operating characteristic curve, AUC > 0.9) in CHOL (0.940), LAML (0.972), KIRC (0.926), OV (0.971), PAAD (0.920), UCEC (0.975), and UCS (0.916) (Figure 3D).

### 2.4. Enrichment Analysis of TRPM2-Related Genes in Pan-Cancer

Correlation analysis was performed using the Pearson correlation to screen out the top 300 genes that were most positively and negatively correlated with TRPM2 in each cancer. Thereafter, we performed Gene Ontology (GO) and Kyoto Encylopaedia of Genes and Genomes (KEGG) analysis of corresponding 301 genes (including TRPM2) in different tumors. We concentrated on OV as an example. The top 50 positively and negatively correlated genes were exhibited in Figure 4A,B, respectively. Interestingly, we found that, in the case of the Biological Process (BP) category of GO, the TRPM2-related genes were enriched in the immune-related activities and pathways, such as “Neutrophil activation involved in immune response”, “Regulation of innate immune response”, “T cell activation”, and “regulation of lymphocyte activation”. (Figure 4C). The KEGG results also suggested that TRPM2-related genes were mainly implicated in pathways regulating immune response and tumorigenesis, such as “Th1 and Th2 cell differentiation”, “Natural killer cell mediated cytotoxicity”, “Antigen processing and presentation”, “HIF−1 signaling pathway”, and “TNF signaling pathway” (Figure 4D). Gene Set Enrichment Analysis (GSEA) results based on the Reactome database were in accordance with the results of GO and KEGG (Figure 4E), indicating that, represented by OV, the TRPM2 was significantly related to various immune-related and tumor-related pathways.

### 2.5. Correlation between TRPM2 Expression and TME-Related Characteristics

From the above gene enrichment analysis, it could be inferred that TRPM2 may participate in cellular immune regulation and tumorigenesis processes. To explore the possible link between TRPM2 and tumor microenvironment, we calculated the TME-related indexes and tumor purity using the ESTIMATE R package. TRPM2 expression was found to be positively linked with stromal, immune, and ESTIMATE scores in the majority of tumors (Figure 5A). Conversely, it had a negative correlation with tumor purity in most cancers. These results were highly suggestive of the important role that TRPM2 plays in tumor microenvironment disorder. We next interrogated the association between TRPM2 expression and the subtypes of infiltrating immune cells using ImmuCellAI, 

TIMER2.0, and EPIC algorithms. Although the results may vary slightly between different algorithms, the overall trend remained the same. The results from ImmuCellAI revealed a positive connection between TRPM2 expression and infiltration score in most tumor types, excluding COAD, DLBC, ESCA, and READ (Figure 5B). The infiltrating of tumor-associated macrophages (TAMs), Tex, Tc, Type 1 T helper (Th1) and immunoregulatory Treg (iTreg) abundances showed a significant positive correlation with TRPM2 expression. In contrast, CD8 naive, Th17, and neutrophil cells were negatively correlated with TRPM2 expression. Meanwhile, there was a significant difference in infiltration score, macrophages, CD8 naive, and iTreg cells between low and high-TRPM2 expression groups in OV patients (Figure 5C–F). Results from TIMER2 supported the positive correlation between TRPM2 expression and the infiltration of TAMs, cancer associated fibroblasts (CAFs), and iTreg cells (Figure 5G). It is worth noting that the correlations between TRPM2 expression and infiltrating levels of TAMs in most tumors were consistent, whichever algorithm was employed (Figure 5B,G,H). We further explored the impact of TRPM2 methylation on immune cell infiltration. As shown in Appendix A, methylation was negatively associated with the infiltration of multiple tumor-infiltrating lymphocytes in most cancers. In particular, in TGCT, the methylation of TRPM2 was negatively associated with infiltration of activated CD8^+^ T cell, activated CD4^+^ T cell, activated B cell, Th1 cell, and natural killer T (NKT) cell. In contrast, it showed a positive correlation with central memory CD8^+^ (Tcm CD8^+^) T cell, Gamma delta T (Tgd) cell, Type 2 T helper (Th2) cell, and immature dendritic (iDC) cell.

### 2.6. Correlations of TRPM2 Expression Levels with Immune Checkpoint Molecules, Immune-Related Genes, and Stemness Score

To estimate the relationship between TRPM2 expression and the potential therapeutic value of the immune checkpoints, we evaluated the association between the expression of TRPM2 and 60 well-known immune checkpoints. Our results suggested a positive connection between TRPM2 and immune checkpoints in most types of cancer, particularly in OV, LIHC, THCA, PAAD, and KIRC. Of note, as demonstrated in Figure 6A, the TRPM2 expression was distinctly related to almost all immune checkpoints (57 out of 60) in OV (Figure 6A). Next, the connection between the expression of TRPM2 and immune activation, immune suppression, chemokines, and their receptors were also analyzed. The results revealed that TRPM2 expression was closely correlated with the level of chemokines and chemokine receptors. For instance, in OV, the expression of TRPM2 was strongly correlated with CC and CXC families, as well as their receptors, such as CCL5, CCL2, CXCL12, CCR5, CCR1 and CCR2 (Figure 6B,C). The results also revealed that there were strong correlations between TRPM2 expression and immune activation genes, and immunosuppressive genes, in almost all malignancies (Appendix A). Furthermore, the stemness of tumor cells was also closely associated with TRPM2: the expression level of TRPM2 was significantly positively correlated to the DNA methylation-based stemness score (DNAss) in 16 tumor types (Figure 6D), including LGG, LUAD, COAD, BRCA, ESCA, STES, KIRP, STAD, PRAD, HNSC, THYM, THCA, READ, PAAD, UVM, and CHOL. Instead, TRPM2 expression exhibited an inverse correlation with DNAss in GBM, UCEC, LIHC, TGCT, and PCPG.

### 2.7. TRPM2 Expression in Different Immune and Molecular Subtypes of Cancers

The relationship of TRPM2 expression with immune and molecular subtypes in pan-cancer was investigated through the TISIDB database. The immune subtypes included the following six categories: C1: wound healing, C2: IFN-γ dominant, C3: inflammatory, C4: lymphocyte depleted, C5: immunologically quiet, C6: TGF-β dominant. As shown in Figure 7 and Figure 8, TRPM2 was differently expressed in diverse immune subtypes of 12 cancer types, including BRCA, CHOL, KICH, KIRC, LGG, LIHC, LUAD, OV, PCPG, SARC, SKCM, and UCEC. Regarding the molecular subtypes, we observed that TRPM2 expression was significantly correlated with different molecular subtypes in 15 tumor types: ACC, BRCA, COAD, ESCA, HNSC, KIRP, LGG, LIHC, LUSC, LUAD, OV, PCPG, PRAD, STAD, SKCM, and UCEC.

### 2.8. Drug Sensitivity Analysis of TRPM2

The correlation between TRPM2 expression and drug sensitivity was analyzed in 809 human cancer cell lines using the GDSC2 database. Results from Spearman correlation analysis revealed that the expression of TRPM2 was negatively correlated with the half maximal inhibitory concentration (IC-50) values of many drugs, among which OSI-027, Acetalax, ML323, VX-11e, Sorafenib, and SCH772984 have the highest negative correlation coefficient (Figure 9).

### 2.9. Expression of TRPM2 at the Single-Cell Level

As can be seen in the distribution heatmap (Figure 10A), we found that TRPM2 was primarily expressed in the Mono/Macro cells cluster in all OV cohorts, notably in GSE147082, GSE154763, and GSE118828. These three cohorts were further analyzed using single-cell scatter plots, which were divided into various types of cells. As expected, TRPM2 was highly expressed in macrophages and malignant cells in the tumor microenvironment, suggesting that TRPM2 may play a role in communication between immune cells and cancer cells.

### 2.10. Expression Profile of TRPM2 in Ovarian Cancer

To estimate the reliability of the above results, we selected OV for further experimental evaluation. The mRNA level of TRPM2 was significantly higher in OV tissues compared with normal tissues (Figure 11A), which was in agreement with the result generated by the TCGA dataset. Our previous analysis revealed that a significant correlation existed between TRPM2 and infiltration of TAMs in a majority of tumors. Therefore, we further detected the expression of CD206, a marker of M2 macrophages, in OV tissues by RT-qPCR to verify their connection. As indicated in Figure 11B, the expression of CD206 in OV tissues highly expressing TRPM2 was upregulated compared with that with lower TRPM2 expression. Correlation analysis revealed a significant positive correlation between two genes. The protein expression level of TRPM2 was subsequently detected in 20 OV tissues and 20 corresponding normal tissues using immunohistochemical staining. The results also confirmed that the expression of TRPM2 was significantly higher in OV compared to normal tissue at the protein level (Figure 11C).

### 2.11. Knockdown of TRPM2 Inhibits the Proliferation in A2780 Cell

After transfecting vector or sh-TRPM2 plasmids in OV A2780 cell, the transfection effectiveness was verified by RT-qPCR. The results displayed in Figure 11D demonstrate that the expression of TRPM2 in the transfected A2780 cell line could be significantly reduced. The proliferative capabilities oPM2 knockdown OV cells were examined through the CCK8 assays at 0 h, 24 h, 48 h, and 72 h of culture. The cell viability was significantly suppressed following the transfection of shRNA-TRPM2 compared with the sh-NC group and the inhibition effect was most pronounced at 72 h of culture (Figure 11E).

### 2.12. Effects of Knockdown TRPM2 on the Apoptosis of A2780 Cell

To observe the effect of TRPM2 on ovarian cancer cell apoptosis, flow cytometry was performed. The apoptosis of A2780 cells with TRPM2 knockdown was detected. As indicated in Figure 11F, compared to the sh-NC group, we observed that knockdown of TRPM2 promoted ovarian cancer cell apoptosis. These findings revealed that knockdown of TRPM2 could suppress some malignant biological behaviors of the ovarian cancer cell and may execute its inhibitory function in the progression of ovarian cancer.

## 3. Discussion

The emergence of evidence has revealed that TRPM2, a highly Ca^2+^ permeable cation channel, acts as a regulator in cancer growth and survival, and is correlated with poor prognosis in patients with breast, gastric, pancreatic, prostate, head and neck cancers, melanoma, and neuroblastoma. In contrast, in a small number of malignancies, the activation of TRPM2 reduces tumor cell survival. The reason for this discordance is likely attributed to the inconsistent carcinogenic mechanisms in the different tumors, which require a comprehensive in-depth analysis. Additionally, while TRPM2 has been studied in certain types of tumors, its role is still elusive in multiple cancers and its association with antitumor immune response remains to be determined.

In the present study, we first explored TRPM2 expression features among pan-cancer and discovered that TRPM2 tended to be highly expressed in almost all TCGA cancers, which indicated that TRPM2 could function as an oncogene for tumor growth and progression. The differential expression of TRPM2 between tumors and corresponding normal tissues suggested that TRPM2 may be helpful in distinguishing tumor patients from healthy individuals. The diagnostic accuracy of TRPM2 in different patients was verified by ROC curve analysis based on TRPM2 expression. Aberrant expression of TRPM2 was validated using OV fresh tissues and paraffin-embedded specimens from our center: when compared with the normal ovarian tissue, an elevated expression of TRPM2 in the OV tissue was found both at the mRNA and protein levels. Furthermore, according to the survival analysis, TRPM2 overexpression was found to act as a risk factor predicting worse prognosis in 15 malignancies. These results underscored the utility of TRPM2 as a potential diagnostic and prognostic biomarker in several tumor types.

Immunotherapy has gradually become an important strategy for tumor treatment and the application of immune checkpoint inhibitors has improved the clinical outcomes of certain tumor patients. The tumor microenvironment profoundly influences tumor progression, metastasis, and effect of chemotherapy and immunotherapy. Understanding tumor immune cell infiltration in TME could facilitate the optimization of antitumor therapy, and targeting of specific cells in the TME has recently attracted much attention in the cancer research community [24]. In the current analysis, we first found that TRPM2 correlated strongly with TME in a majority of tumors by evaluating the relationship between TRPM2 expression and three TME-related scores, i.e., Stromal, Immune, and Estimate Score, as well as tumor purity. TRPM2 was also closely associated with the infiltration of multiple immune cells in a broad range of malignancies, and in particular positively correlated with TAM infiltration in all three distinct algorithms. We further interrogated the distribution of TRPM2 at a single-cell level and verified the main expression of TRPM2 at macrophages and malignant cells in the TME. Not only that, but the RT-qPCR results from clinical specimens also validated the upregulated TRPM2 expression and positive correlation between the expression level of TRPM2 and M2 macrophage marker CD206 in ovarian cancer. TAMs comprise a major proportion of TILs in the TME [25] and normally maintain the functional properties of M2 macrophages [26]. Most TAMs not only lose the ability to fight against tumor progression but also orchestrate the construction of a dysregulated TME via the secretion of diverse immunosuppressive cytokines. In addition, oxidative stress is an additional metabolic feature in the TME [27]. ROS located in the TME is an important regulator of tumorigenesis, cell proliferation, apoptosis, and therapeutic response [28]. The overexpressed TRPM2 channel could be activated by ROS-induced Ca^2+^entry, which in turn boosts activation of this channel, leading to a positive feedback mechanism and increased oxidative stress [29], ultimately leading to a more malignant and refractory TME. Our above results indicated that TRPM2 may play a role in an immune-dysregulated TME by affecting immune cell infiltration and communication between immune and cancer cells. In addition, we infer that TRPM2 could be explored as a therapeutic target with the potential to aid in the functional suppression and even the repolarization of M2 TAMs.

Apart from the type and number of TILs, the expression of immune checkpoint genes on tumor cells, such as PD-L1 and CTLA-4, have been reported as predictive biomarkers for the therapeutic efficacy of immune checkpoint inhibitors in some cancers [30,31]. In probing the association between TRPM2 and checkpoint gene expression, we found that TRPM2 was relatively correlated with the expressions of checkpoint genes in most cancers and this phenomenon was more prominent in OV, with 57 out of 60 checkpoint genes’ expression significantly correlated with TRPM2. This suggested that the TRPM2 may be used as a marker to predict the effect of immunotherapy, particularly for patients with ovarian cancer. Moreover, the strong positive correlations across the immune checkpoint genes indicated that TRPM2 may be able to form positive feedback with certain immune checkpoint genes, facilitating the immune escape of tumor cells. These mechanisms also highlight the predictive and therapeutic values of TRPM2 for tumor precision therapy.

The correlation between TRPM2 and immune-related genes, including chemokine and their receptors, and immunoregulatory genes, similarly demonstrated its intimate interactions with tumor immunity. Chemokines and chemokine receptors play crucial roles in mediating immune cell trafficking into the TME [32], which is required to initiate and deliver effective antitumor immune responses. Chemokine secretion in the TME is often altered, and aberrant chemokine profiles can promote the differentiation and infiltration of immunosuppressive tumor-promoting cells (i.e., Treg cells, MDSCs, and TAMs) into tumors [33]. Our prior analysis revealed that the infiltration of TAMs was strongly correlated with the TRPM2 expression level in most tumors. What is known is that CCL2/CCR2 axis is important for the recruitment of TAMs [34], CCR5 binds both to CCL3 and CCL5, in turn recruiting myeloid cells.CCR5 blockade can repolarize TAMs from a pro-tumorigenic phenotype to an M1 phenotype [35]. We thus speculated whether the expression of TRPM2 is related to the chemokines mentioned above. As expected, TRPM2 was significantly positively correlated to most chemokines and chemokine receptors. In addition, in the OV cohort, TRPM2 was also associated with several chemokines and chemokine receptors that have been shown to promote ovarian cancer progression and metastasis, such as CXCL11-CXCR3 and CXCL12-CXCR4 chemokine axes [36,37].

Tumor progression involves the gradual loss of differentiated phenotypes and the acquisition of stemness characteristics, which have been confirmed to represent the capability of tumor proliferation and contribute to bad outcomes. Our analysis revealed that TRPM2 was closely correlated with stromal score and was positively related to cancer cell stemness index in 16 tumor types. This indicates that, among these cancer types, tumor cells with high TRPM2 expression may have a stronger capacity for self-renewal and proliferation compared to those with low TRPM2 expression, and TRPM2 may promote tumor aggressiveness by enhancing the stemness of tumors cell.

In view of the potential impacts of TRPM2 on prognosis and immunoregulation, we further analyzed TRPM2 expression in different molecular and immune subtypes across pan-cancer. The results closely followed those from previous analyses: TRPM2 expression was significantly different in different immune and molecular subtypes in most cancers, which reflected the biomarker potential of TRPM2 in subtyping and guiding tumor therapeutic strategies.

Lastly, we screened out several drugs that target TRPM2 and could act as antitumor agents. Further in vitro experiments were also conducted to investigate the effect of TRPM2 on cell behavior. After knocking down the TRPM2 expression by shRNA plasmid in the A2780 cell line, a significant inhibition of proliferation and promotion of apoptosis was found compared with that in the negative control group. Therefore, these results also verified the cancer-promoting properties of TRPM2 in ovarian cancer.

Our study first comprehensively elucidates that TRPM2 may function as a promising diagnostic and prognostic biomarker. It is also expected to serve as an encouraging therapeutic target for tumor immunotherapy in a variety of tumor types, especially in OV, which provides new insight into tumor precision medicine.

Unfortunately, there were still some shortcomings in this study. First, although we used the clinical specimens to verify part of our bioinformatic analysis results, the sample size of this research was relatively small, and more clinical research and data are needed. Secondly, though the immunomodulatory role of TRPM2 has been explained theoretically, the specific mechanisms have not been clarified, and this requires investigations in depth at a later stage. Moreover, further studies should be carried out to implement targeting of TRPM2 as an anti-cancer therapeutic strategy.

## 4. Materials and Methods

### 4.1. Acquiring and Analysis of TRPM2 Expression Data

The Gene expression, somatic mutations, and corresponding clinical data among 33 tumor types were retrieved from The Cancer Genome Atlas (TCGA) database (https://portal.gdc.cancer.gov/, accessed on 5 November 2022). Meanwhile, RNA expression data of 26 types of normal tissues came from the Genotype-Tissue Expression (GTEx) databases (https://www.gtexportal.org/home/index.html, accessed on 7 November 2022). The Cancer Cell Line Encyclopedia (CCLE) database (https://portals.broadinstitute.org/ccle, accessed on 7 November 2022) was used to derive gene expression data in various tumor cell lines.

### 4.2. Diagnostic and Prognostic Analysis

To investigate the prognostic value of TRPM2 in various cancers, we explored the connection between TRPM2 expression and survival outcomes, including the overall survival (OS), disease-free interval (DFI), progression-free interval (PFI), and disease-specific survival (DSS), of patients according to data from TCGA database. R packages “limma”, “survival”, and “survminer” were used to perform Kaplan–Meier and univariate Cox proportional-hazards analyses and visualize the hazard ratio (HR) and 95% confidence intervals (95% CI). Furthermore, the clinical data obtained from TCGA was used to assess the correlation between TRPM2 expression and pathological stages. The R package “pROC” was utilized to perform receiver operating characteristic curve analysis to explore TRPM2 predicted values in TCGA tumor tissues and the values in the matching normal tissues, with AUC > 0.9 indicating good diagnostic ability.

### 4.3. Genetic Alteration Analysis

The cBioPortal web (https://www.cbioportal.org/, accessed on 12 November 2022) was used to analyze and visualize the genetic alterations, which include genetic alteration frequency, mutation type, and copy number alteration (CNA) of TRPM2 in pan-cancer. The associations between CNA, DNA methylation, and expression level of TRPM2 among different tumors were explored using Pearson correlation analysis. The gene mutation profiles of high and low TRPM2 expression groups were plotted using the “maptools” R package.

### 4.4. Functional Enrichment Analysis

To appraise the potential biological process and downstream pathways in which TRPM2 is involved, we screened out the top 100 genes which have the most significant positive and negative correlation with TRPM2 to form a gene set. Next, R packages “clusterProfiler” and “ggplot2” were applied to perform gene ontology (GO), and Kyoto Encyclopedia of Genes and Genes (KEGG) analyses. Furthermore, A gene set enrichment analysis (GSEA) analysis was performed based on the “Reactome” database to further explore the potential pathways that TRPM2 involved.

### 4.5. Correlations between TRPM2 Expression and Immunity

For all tumor samples, Immune, Stromal, and ESTIMATE scores, which represent the proportion of immune cells and stromal cells in the TME, were calculated to predict tumor purity using the “limma” and “ESTIMATE” R packages. To evaluate the tumor stemness index, we obtained the DNAss calculated by machine learning in previous studies [38] and performed Pearson’s correlation analysis. We additionally determined the correlation between the immune cell infiltration and the expression of TRPM2 using the data from TIMER 2.0 (http://timer.comp-genomics.org/, accessed on 17 November 2022) [39], ImmuCellAI (http://bioinfo.life.hust.edu.cn/ImmuCellAI#!/, accessed on 17 November 2022), and EPIC algorithm by “IOBR” R package. The effect of TRPM2 methylation on 28 tumor-infiltrating lymphocytes (TILs) types was analyzed based on immune-related signatures from Charoentong’s study [40]. We explored the effects of TRPM2 on immune regulation by investigating the correlation between TRPM2 expression and immune-related genes, including immune checkpoint genes, immune suppression genes, chemokines, and chemokine receptors. The correlation between TRPM2 expression levels and immune or molecular subtypes of tumor types was further explored using data from the TISIDB database (http://cis.hku.hk/TISIDB/, accessed on 22 February 2023) [41].

### 4.6. TRPM2 Expression at Single Cell Level

To uncover the distribution of TRPM2 at the single cell level, the expression of TRPM2 in TME-related cells, including immune cells, stromal cells, malignant cells, and functional cells, was analyzed using the Tumor Immune Single-cell Hub 2 (TISCH2) database (http://tisch.comp-genomics.org/home/, accessed on 19 April 2023).

### 4.7. Drug Sensitivity Analysis

The GDSC2 database (https://www.cancerrxgene.org/, accessed on 5 June 2021) contains molecular and pharmacological datasets for the 809 different human tumor cell lines from nine cancer types and 198 compounds. Relationships between TRPM2 mRNA expression and IC50 of diverse drugs were investigated using Pearson correlations.

### 4.8. Clinical Tissue Specimen Acquisition

All tissue samples, including 20 OV and 20 normal ovarian tissues, were collected from surgical resections at the Department of Gynecology, the West China Secondary Hospital. Tumor samples were extracted from patients with primary OV and verified by operative pathology. None of the OV patients had previously undergone chemo-, radio-, or immunotherapy before their operations. Normal ovarian tissues were obtained from patients with benign diseases undergoing hysterectomy and bilateral salpingectomy. After surgical removal, specimens were immediately snap-frozen in liquid nitrogen and stored at -80° C. All patients provided informed consent and approval of the Ethical Review Board of West China Secondary Hospital was also acquired.

### 4.9. Immunohistochemical (IHC) Experiment

We collected 20 pairs of formalin-fixed, paraffin-embedded (FFPE) archival blocks of OV tissues and corresponding normal ovary tissues from West China Secondary Hospital. After deparaffinization and dehydration of the tissue sections, antigen retrieval was performed in citrate buffer using the microwave, followed by incubation with 3% H_2_O_2_. Sections were then incubated with anti-human TRPM2 antibody overnight at 4 °C (1/100 dilution, HuaBio, Hangzhou, China), HA500437). Subsequently, we conjugated the sections with secondary antibody at room temperature for 2 h. Diaminobenzidine (DAB) kit (Solarbio, Beijing, China, DA-1010) was used to develop color. Finally, 3–5 viewing fields were randomly chosen under light microscopy. The color intensity of positive cells ranges from 0 to 3, with 0 being weak and 3 being intense staining. The staining range was scored as follow: 0 represented <5%, 1 represented 5–25%, 2 represented 26–50%, 3 represented 51–75%, and 4 represented 76–100%. The IHC staining score was calculated by multiplying the intensity score and staining range and a score greater than 2 was defined as positive staining.

### 4.10. Cell Culture and Transfection

Ovarian cancer cell line (A2780) was cultured in high-glucose Dulbecco’s modified Eagle’s medium (DMEM; Bio-Channel, Shanghai, China) containing 10% fetal bovine serum (FBS; Gibco, Grand Island, NY, USA) and 1% penicillin-streptomycin (Beyotime, Shanghai, China), and incubated at 37 °C in a humidified incubator. Sh-TRPM2 or negative control (sh-NC) plasmids (Sangon, Shanghai, China) were transfected into cells using Lipofectamine 3000 (Life Technologies Corporation; Thermo Fisher Scientific, Inc., Waltham, MA, USA), according to the recommended protocol. After 48 h, the stably transfected cells were screened out using puromycin (6 μg/mL) to conduct subsequent experiments.

### 4.11. RNA Extraction and Real-Time Quantitative PCR (RT-qPCR)

Total RNA from the tissue specimens and cultured cells was extracted using TRIzol (CWBio; Beijing, China) according to the manufacturer’s protocol. Next, 1 μg of total RNA was reverse-transcribed into cDNA using the Hifair II 1st Strand cDNA Synthesis Kit (Yeason, Shanghai, China), and Hieff UNICON Power qPCR SYBR Green Master Mix (Yeason, Shanghai, China) was used for RT-qPCR. All steps were performed on ice. The relative RNA expression level was calculated using the 2^−ΔΔCT^ method with GAPDH as an endogenous control.

### 4.12. Cell Counting Kit-8 (CCK-8) Assay

After puromycin selection, the cells were seeded into 96-well plates at a density of 5000 cells per well (100 μL medium) in triplicate for each group and cultured for 0 h, 24 h, 48 h, and 72 h. At the indicated time points, 10 μL of the CCK-8 solution (Yeason, Shanghai, China) was added to each well and incubation was continued for 2 h at 37 °C, 5% CO2. The optical density (OD) was measured through a microplate reader (Biotek Instruments, Inc., Winooski, WT, USA) at 450 nm.

### 4.13. Cell Apoptosis Detection

For the analysis of apoptosis, an Annexin APC-FITC/7AAD Detection Kit (Elabscience, Wuhan, China) was used according to the manufacturers’ instructions to label apoptotic cells. Cells were seeded in 6-well plates (1 × 10^6^/well) to 90% confluence, harvested, and then incubated at room temperature in the dark after Annexin V-APC and/or 7AAD was added. The expression of Annexin V-APC and 7AAD staining was detected using flow cytometry.

### 4.14. Statistical Analysis

All analyses and mapping were performed using R language (version 4.0.3) and GraphPad Prism (version 8.0) software. The Wilcoxon rank-sum test was used to test differences in expression between tumor and normal tissues. Spearman or Pearson correlation analysis was employed to calculate the correlation. The *p* value of less than 0.05 was considered to be statistically significant and * *p* < 0.05, ** *p* < 0.01, *** *p* < 0.001.

## Figures and Tables

**Figure 1 ijms-24-11912-f001:**
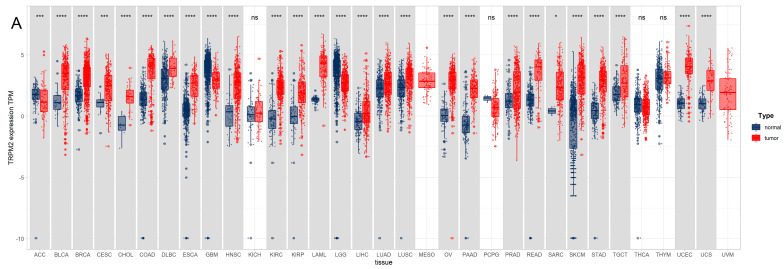
Expression profile of Transient receptor potential melastatin 2 (TRPM2) across different tumor types. (**A**) The expression level of TRPM2 in different tumors compared to normal tissues; (**B**) the relative expression of TRPM2 in pan-cancer; (**C**) differential expression of TRPM2 between paired tumor and adjacent normal tissues; (**D**) TRPM2 expression in different human cancer cell lines. The red dot represents the average expression value of TRPM2 in different cell lines, while the blue and yellow lines represent the scale, similar to the *y*-axis scale. ns represents non-significant; * indicates *p*-value < 0.05; ** *p* < 0.01; *** *p* < 0.001; **** *p* < 0.0001.

**Figure 2 ijms-24-11912-f002:**
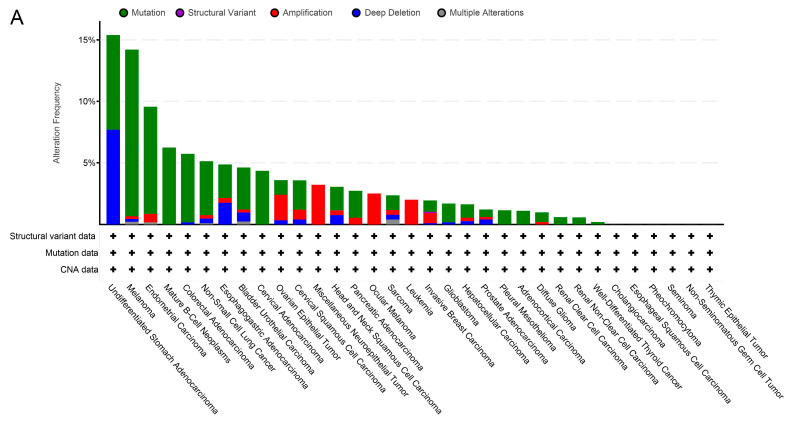
The genetic alterations of TRPM2 across cancer types. (**A**) Alterations summary of TRPM2 in pan-cancer set; (**B**) relationship between copy number alteration (CNA) and gene expression; (**C**) relationship between DNA methylation and TRPM2 expression.

**Figure 3 ijms-24-11912-f003:**
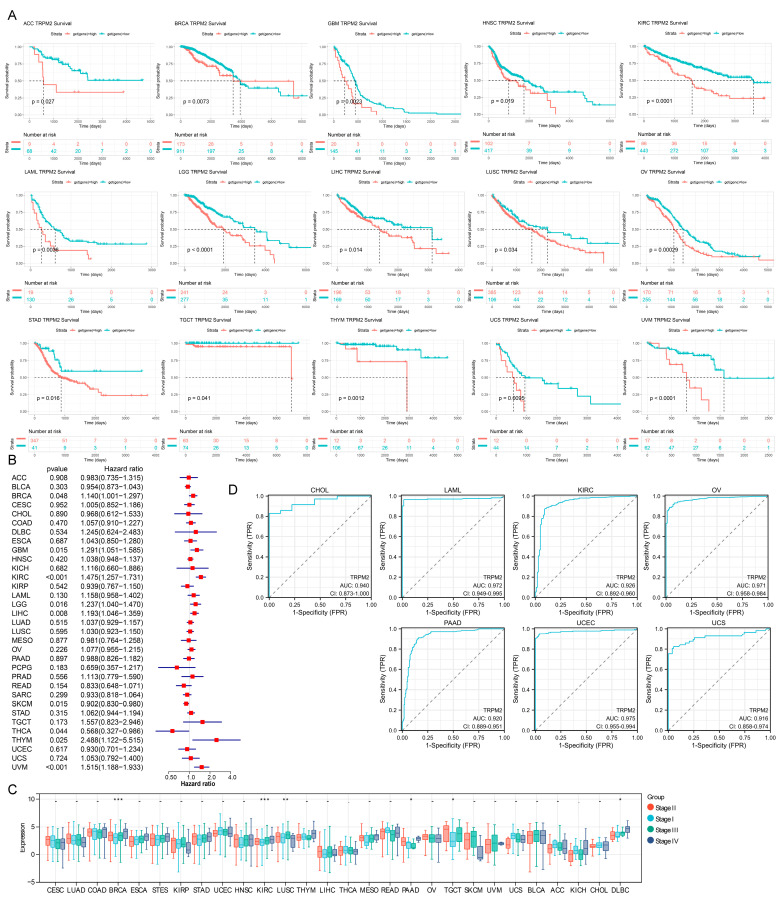
Diagnostic and prognostic analysis of TRPM2 in pan−cancer. (**A**) Comparison of overall survival (OS) curves between high and low expression of TRPM2 groups in some cancer types by Kaplan Meier methodology. Red line represents high TRPM2 expression group and blue line represents low expression group; (**B**) relationship between TRPM2 expression and OS using the Cox regression analysis in pan-cancer; (**C**) expression levels of TRPM2 at distinct clinical stages in pan-cancer. * *p* < 0.05; ** *p* < 0.01; *** *p* < 0.001; (**D**) receiver operating characteristic (ROC) curves of TRPM2 in some tumor types with area under the ROC curve (AUC) > 0.90.

**Figure 4 ijms-24-11912-f004:**
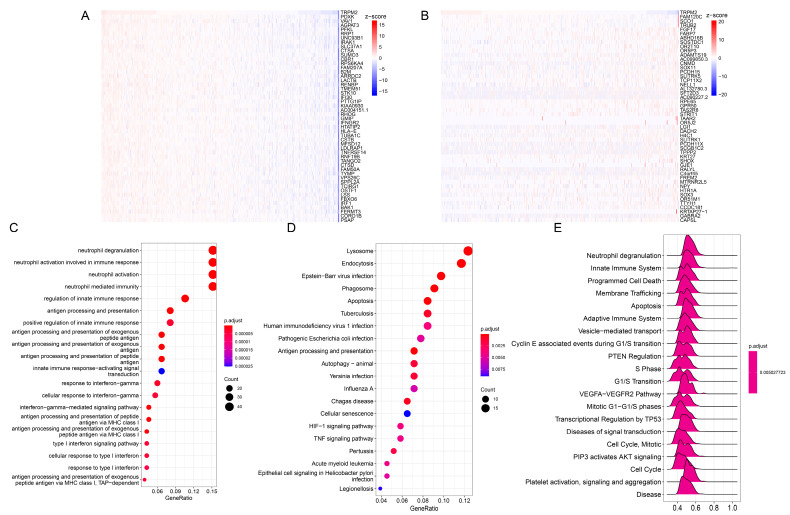
Pathway enrichment analysis of TRPM2-related genes in ovarian cancer (OV). (**A**,**B**) Heatmaps of the top 50 genes that were positively associated (**A**) and the top 50 genes that were negatively associated (**B**) with TRPM2 expression; (**C**) results of Gene Ontology enrichment analysis in biological process category (the size of the dots represent the count—the redder the color of the dot, the smaller the *p*-value); (**D**) the top 20 pathways correlated with TRPM2 according to Kyoto Encyclopedia of Genes and Genomes (KEGG) pathway enrichment analysis; (**E**) results of Gene Set Enrichment Analysis (GSEA) based on the Reactome database.

**Figure 5 ijms-24-11912-f005:**
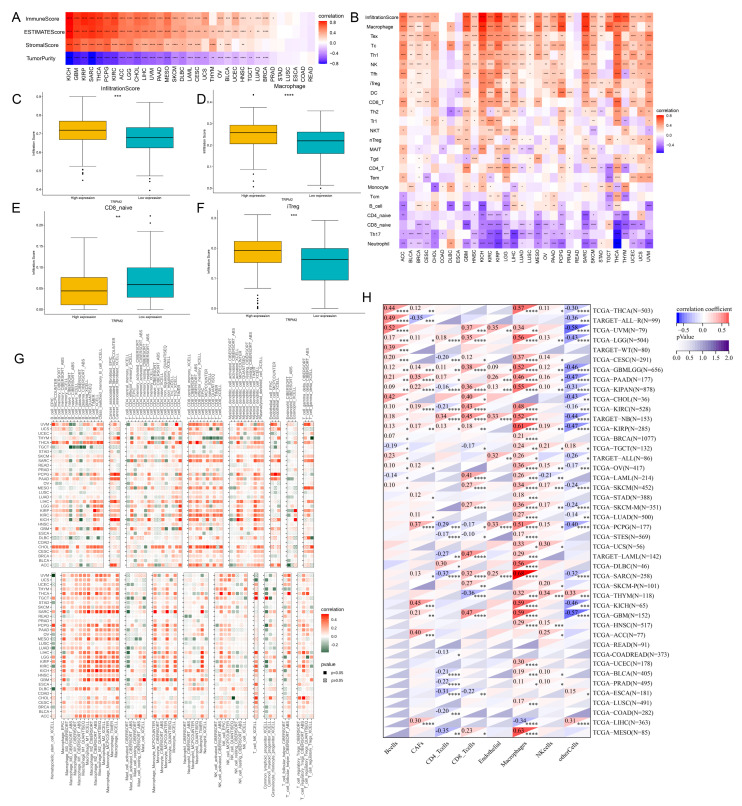
The correlation between TRPM2 expression and immunocyte infiltration in human pan-cancer. (**A**) Correlation of TRPM2 expression with three TME-related scores (stromal, immune, and ESTIMATE score) and tumor purity; (**B**) analysis of immune cell infiltration based on ImmuCellAI database (different shades of red color represent positive correlations, and different shades of blue represent negative correlations); (**C**–**F**) differences in infiltration score, infiltration of macrophages, CD8 naive, and iTreg cells between low and high-TRPM2 expression group in OV patients, respectively (yellow represents high expression and green represents low expression); (**G**) analysis of immune cell infiltration based on TIMER2.0 database (different shades of red represent positive correlations, and different shades of green represent negative correlations); (**H**) analysis of immune cell infiltration based on EPIC algorithm. * *p* < 0.05; ** *p* < 0.01; *** *p* < 0.001; **** *p* < 0.0001.

**Figure 6 ijms-24-11912-f006:**
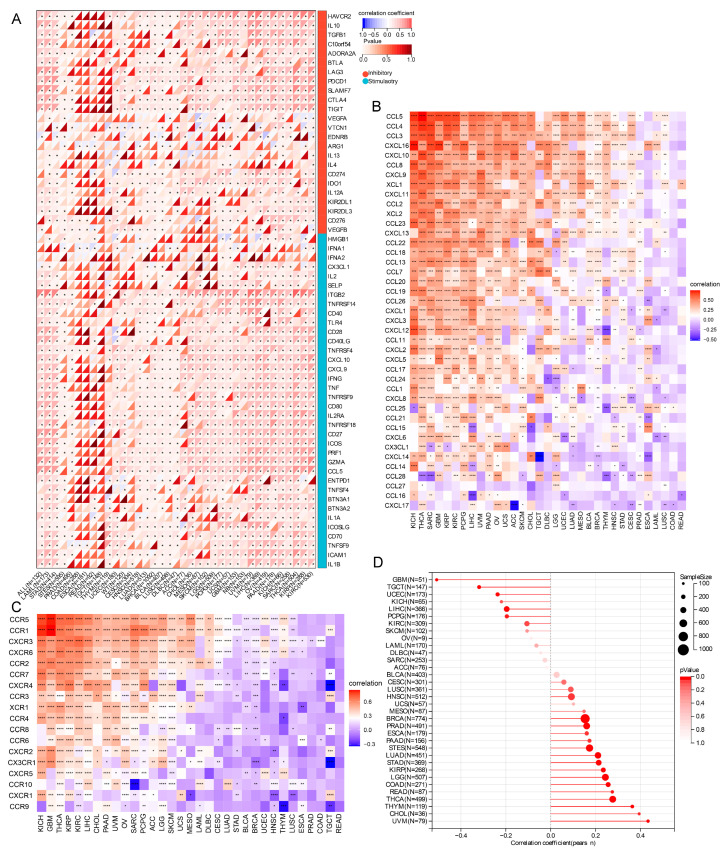
Pan-cancer correlation analysis of TRPM2 expression and immune-related signature. (**A**) Relationship between TRPM2 expression and 60 immune checkpoint genes (the color of the upper left triangle represents correlation, with red representing positive correlation and blue representing negative correlation—different shades of red in the lower right triangle represent *p* value, asterisks indicate significant correlations); (**B**,**C**) relationship between TRPM2 expression and chemokines (**B**) and chemokine receptors (**C**), respectively (different shades of red represent positive associations, and different shades of blue represent negative associations). * *p* < 0.05; ** *p* < 0.01; *** *p* < 0.001; **** *p* < 0.0001; (**D**) lollipop graph of correlations between TRPM2 expression and stemness scores. The size of the dots represent the sample size. The redder the dot, the smaller the *p*-value.

**Figure 7 ijms-24-11912-f007:**
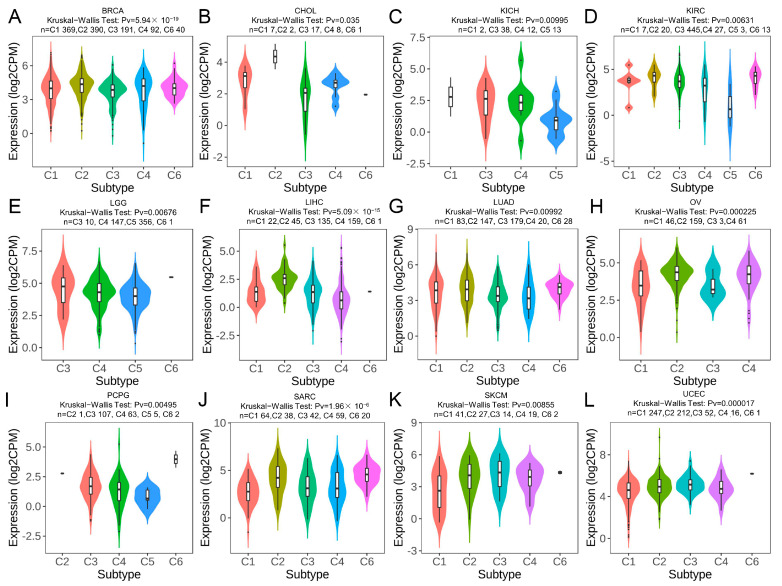
TRPM2 expression in pan-cancer immune subtypes. TRPM2 expression in immune subtypes of (**A**) BRCA; (**B**) CHOL; (**C**) KICH; (**D**) KIRC; (**E**) LGG; (**F**) LIHC; (**G**) LUAD; (**H**) OV; (**I**) PCPG; (**J**) SARC; (**K**) SKCM; and (**L**) UCEC. The abscissa represents various subtypes of different tumors, and the ordinate represents the TRPM2 expression.

**Figure 8 ijms-24-11912-f008:**
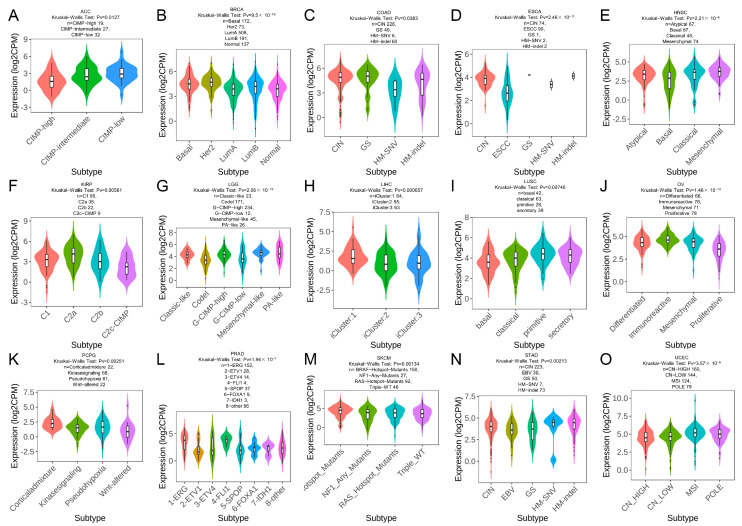
TRPM2 expression in pan-cancer molecular subtypes. TRPM2 expression in molecular subtypes of (**A**) ACC; (**B**) BRCA; (**C**) COAD; (**D**) ESCA; (**E**) HNSC; (**F**) KIRP; (**G**) LGG; (**H**) LIHC; (**I**) LUSC; (**J**) OV; (**K**) PCPG; (**L**) PRAD; (**M**) SKCM; (**N**) STAD; and (**O**) UCEC. The abscissa represents various subtypes of different tumors, and the ordinate represents the TRPM2 expression.

**Figure 9 ijms-24-11912-f009:**
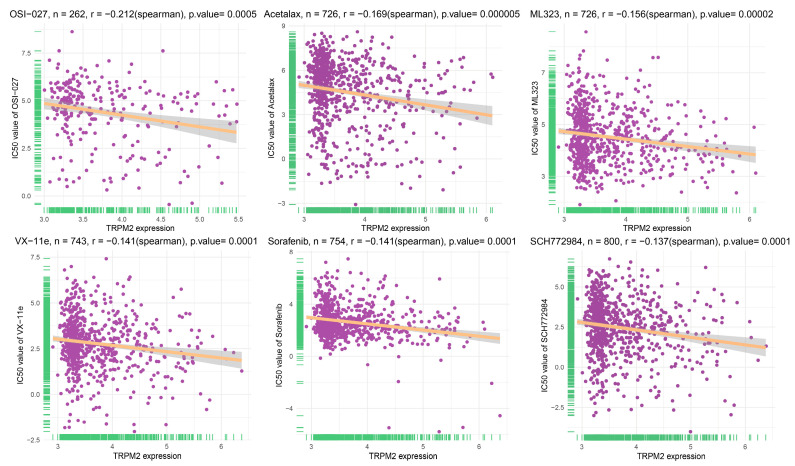
The correlation between TRPM2 expression and the half maximal inhibitory concentration (IC-50) values of drugs.

**Figure 10 ijms-24-11912-f010:**
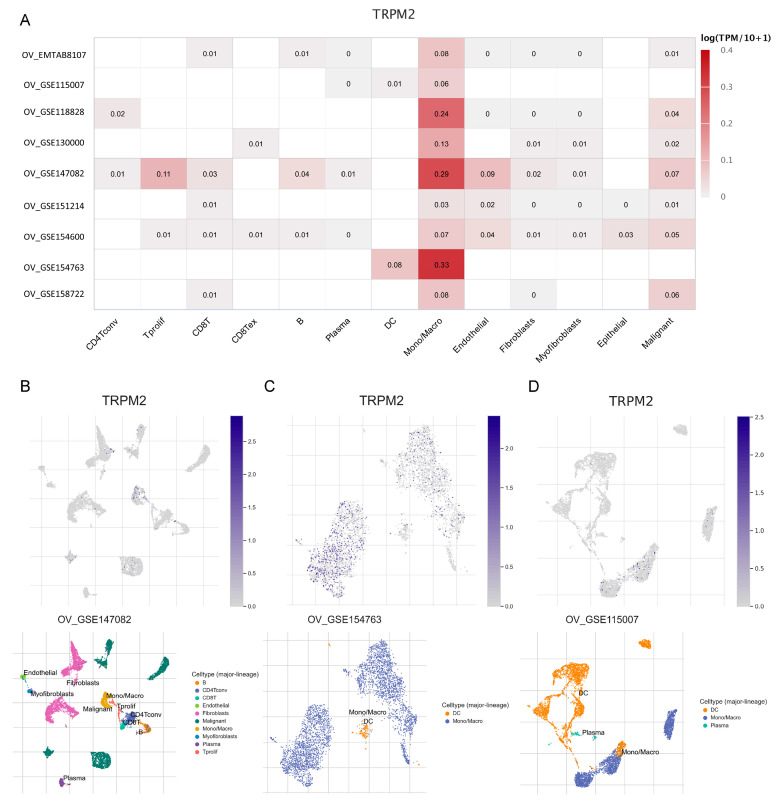
Expression of TRPM2 at the single-cell level. (**A**) Summary of TRPM2 expression of 13 cell types in OV single cell databases; (**B**–**D**) scatter plot demonstrating the distributions of different cell types (**Lower**) and the TRPM2 expression levels (**Upper**) in the GSE147082, GSE154763, and GSE118828 database.

**Figure 11 ijms-24-11912-f011:**
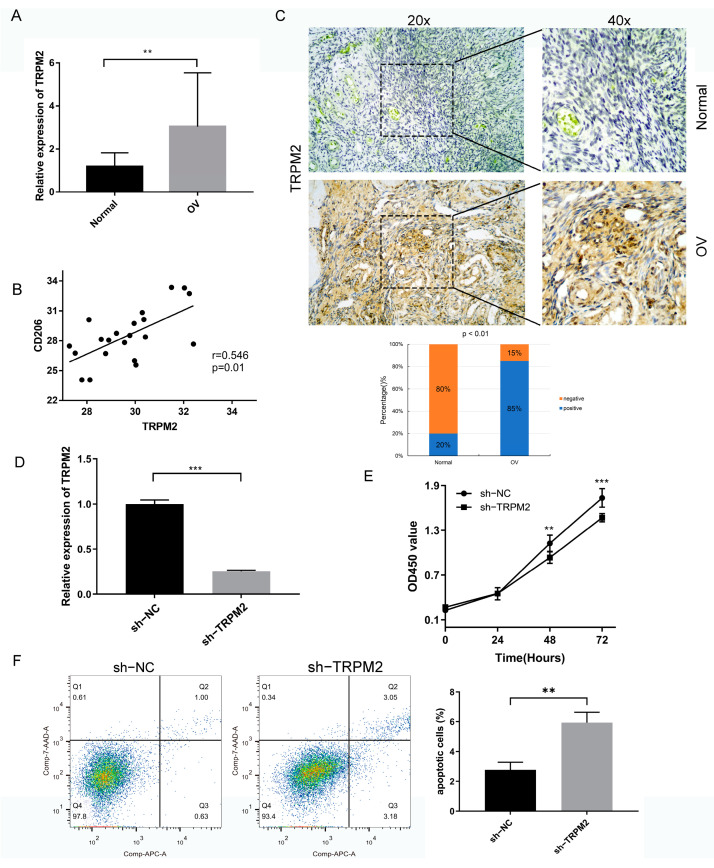
Expression validation in ovarian cancer (OV) and effect of sh-TRPM2 on the proliferation and apoptosis abilities of ovarian cancer cell A2780. (**A**) Validation of the different mRNA expression level of TRPM2 between normal tissues and OV tissues using the RT-qPCR; (**B**) correlation of TRPM2 expression with M2 macrophages marker CD206 expression in OV; (**C**) representative immunohistochemical staining of TRPM2 protein in OV tissue and corresponding normal tissue (the Histogram shows the percentage of positive or negative staining in normal and OV tissues); (**D**) the effect of the knockdown was validated through RT-qPCR; (**E**) CCK-8 assay to detect the proliferative capacity; (**F**) apoptosis of A2780 cells was assessed by flow cytometry. All the experiments were replicated at least three times. Data were expressed as the mean  ±  s.d. ** *p* < 0.01; *** *p*  <  0.001.

## Data Availability

The public available databases analyzed during the current study are included in the article, further inquiries can be directed to the corresponding author on reasonable request.

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
