# Peer review of "Identification of TRPM2 as a Potential Therapeutic Target Associated with Immune Infiltration: A Comprehensive Pan-Cancer Analysis and Experimental Verification in Ovarian Cancer"

_ijms, 2023, doi:10.3390/ijms241511912_

Round 1

Reviewer 1 Report

Danxi Zheng et al. reported that TRPM2 is a potential therapeutic target associated with immune infiltration in ovarian cancer. The authors well explained the rationale for TRPM2 as a potential therapeutic target in OV with convincing data.

Some of the comments are enclosed here to improve the quality of the manuscript.

1.     In Figure 1C, the significance (P Value) is completely not visible.

2.     Some the of figure legends need to be described more, especially Fig 6 and Fig 7.

3.     In the 2.10 section, “The mRNA level of TRPM2 was significantly higher in OV tissues com-289 pared with normal tissue (Figure.11A),” Is it primary or metastatic OV tissue or What is the mRNA expression of TRPM2 in primary Vs Metastasis OV tissues.

4.     In the abstract, line 25 “The overexpressed TRPM2 in OV tissues was validated at both the mRNA and protein levels” but the protein levels did not show anywhere in the manuscript.

5.     In Figure 11D, it is better to keep sh-NC first (left side) and then sh-TRPM2 (right side) in the graph which is the most acceptable way of graphical representation.

6.     What is the P value (statistical significance) in Figures 11D, E, and F also please mention in the figure legend how many repeats of each experiment.

Reviewer 2 Report

The authors have provided sufficient data to establish the role of TRPM2 in cancer.

 However, the data presented in this study requires improvement in terms of figure visibility to facilitate comprehension by readers.

The authors missed the published article to include in the current manuscript “TRPM2-mediated Ca2+ signaling as a potential therapeutic target in cancer treatment: an updated review of its role in survival and proliferation of cancer cells (PMID: 37337283)”.

Line 522: Remove the extra dot from the middle of a sentence.
